# *Terminalia bentzoë*, a Mascarene Endemic Plant, Inhibits Human Hepatocellular Carcinoma Cells Growth In Vitro via G0/G1 Phase Cell Cycle Arrest

**DOI:** 10.3390/ph13100303

**Published:** 2020-10-12

**Authors:** Nawraj Rummun, Philippe Rondeau, Emmanuel Bourdon, Elisabete Pires, James McCullagh, Timothy D. W. Claridge, Theeshan Bahorun, Wen-Wu Li, Vidushi S. Neergheen

**Affiliations:** 1Department of Health Sciences, Faculty of Medicine and Health Sciences, University of Mauritius, Réduit 80837, Mauritius; n.rajeevr10@gmail.com; 2Biopharmaceutical Unit Centre for Biomedical and Biomaterials Research, MSIRI Building, University of Mauritius, Réduit 80837, Mauritius; tbahorun@uom.ac.mu; 3School of Pharmacy and Bioengineering, Faculty of Medicine and Health Sciences, Keele University, Thornburrow Drive, Stoke on Trent ST4 7QB, UK; 4Université de La Réunion, INSERM, UMR 1188 Diabète athérothrombose Thérapies Réunion Océan Indien (DéTROI), Saint-Denis de La Réunion, 97490 Sainte-Clotilde, Reunion, France; rophil@univ-reunion.fr (P.R.); emmanuel.bourdon@univ-reunion.fr (E.B.); 5Chemical Research Laboratory, University of Oxford, Oxford OX1 3TA, UK; elisabete.pires@chem.ox.ac.uk (E.P.); james.mccullagh@chem.ox.ac.uk (J.M.); tim.claridge@chem.ox.ac.uk (T.D.W.C.)

**Keywords:** *Terminalia bentzoë*, Mascarene endemic, cancer, cytotoxicity, antioxidant, cell cycle arrest, phenolics, bioassay-guided fractionation

## Abstract

Tropical forests constitute a prolific sanctuary of unique floral diversity and potential medicinal sources, however, many of them remain unexplored. The scarcity of rigorous scientific data on the surviving Mascarene endemic taxa renders bioprospecting of this untapped resource of utmost importance. Thus, in view of valorizing the native resource, this study has as its objective to investigate the bioactivities of endemic leaf extracts. Herein, seven Mascarene endemic plants leaves were extracted and evaluated for their in vitro antioxidant properties and antiproliferative effects on a panel of cancer cell lines, using methyl thiazolyl diphenyl-tetrazolium bromide (MTT) and clonogenic cell survival assays. Flow cytometry and comet assay were used to investigate the cell cycle and DNA damaging effects, respectively. Bioassay guided-fractionation coupled with liquid chromatography mass spectrometry (MS), gas chromatography-MS, and nuclear magnetic resonance spectroscopic analysis were used to identify the bioactive compounds. Among the seven plants tested, *Terminalia*
*bentzoë* was comparatively the most potent antioxidant extract, with significantly (*p* < 0.05) higher cytotoxic activities. *T. bentzoë* extract further selectively suppressed the growth of human hepatocellular carcinoma cells and significantly halted the cell cycle progression in the G0/G1 phase, decreased the cells’ replicative potential and induced significant DNA damage. In total, 10 phenolic compounds, including punicalagin and ellagic acid, were identified and likely contributed to the extract’s potent antioxidant and cytotoxic activities. These results established a promising basis for further in-depth investigations into the potential use of *T. bentzoë* as a supportive therapy in cancer management.

## 1. Introduction

The plant kingdom is known to be a prolific sanctuary of phytochemicals with unique therapeutic potential. At least 25% of the 1562 clinical drugs approved by the US Food and Drugs Administration are known to have been derived from terrestrial plants [1,2]. Moreover, an estimated 28,187 plant taxa, globally, are documented to have medicinal values, with over 3000 species reported with the ethnomedicinal application against cancer [3,4]. The continued dependence of mankind on plants was further evidenced during the recent outbreak of the COVID-19 pandemic, whereby herbal medicines were used in an attempt to mitigate the symptoms of the novel coronavirus infection [5,6,7].

Madagascar and its neighboring islands in the Western Indian Ocean regions are known as biodiversity hotspots [8]. Undoubtedly, untapped endemic plant species from these niche areas broaden the structural variation of novel chemotypes [9,10]. The tropical forest of Madagascar was once acknowledged as a fertile source of economically valuable plants with pharmacologically active ingredients [11]. Indeed, the anticancer drugs vinblastine and vincristine were derived from the Madagascan endemic *Catharanthus roseus* (Apocynaceae) [12]. Certainly, the market value of vincristine alone was estimated to be USD 15 million per kilogram, in the year 2016 [13].

Phytogeography investigations revealed that the Mascarene endemic plants have their ancestral lineages tracing back to Madagascar [14]. As such, the unique floral biodiversity of Mauritius is expected to possess similar medicinal and therapeutic prolificacy as the Madagascan rain forest. However, instead of conserving such valuable biodiversity, human activities are pushing endemic taxa towards an unprecedented extinction crisis. In less than 400 years of human settlement, Mauritius has witnessed the shrinking of its native forest to around 5% of the original cover, leading to the permanent loss of 30 (10.9%) of its endemic plant species and driving 81.7% of the remnant endemic taxa to the brink of extinction [15,16,17]. For instance, fewer than 500 adults trees of *Terminalia bentzoë* (L.) L.f. subsp. *bentzoë* are recorded in the wild, defining the species survival as vulnerable, as per the International Union for Conservation of Nature red list criteria [18]. Nevertheless, the remnant areas of the pristine forest are still home to a plethora of endemic flora rich in high genetic diversity, representing interesting sources for complementary and alternative medicine, nutraceuticals, as well as pharmaceutical leads [16].

The initiation and progression of cancer involve oxidative stress via DNA damage, and the increase of DNA mutations. Conventional chemo- and radiotherapy cause cancer cell death often through the generation of reactive oxygen species (ROS), but also unfortunately lead to severe side effects. It is highly desired to develop more effective therapies with less toxic effects [19]. Plant polyphenols exert anticarcinogenic activity by interfering with the different hallmarks of cancer, including sustained tumor cell proliferation, angiogenesis and apoptotic cell death, through various signaling pathways [19,20]. Polyphenolics can also behave as antioxidants, thereby maintaining the integrity of DNA from oxidative stress attack and preventing the initiation stage of carcinogenesis [21,22].

The tropical island of Mauritius is known for its endemic biodiversity richness [8]. However, human activities on the island have provoked the irreversible loss of a considerable fraction of this genetic resource. For instance, in less than four centuries, Mauritius has lost 95% of its pristine forest cover, accounting for the extinction of 10.9% of the island’s endemic flora [23]. The scarcity of rigorous scientific data on the surviving endemic taxa renders the bioprospecting of this untapped resource of utmost importance. With this in mind, and in view of providing solid foundations to enforce stringent conservation policies, the in vitro antioxidant propensities of leaf extracts from seven plants endemic to Mascarene islands were investigated, in relation to their polyphenolic content. The plants under study have documented traditional uses against ailments ranging from dermatological conditions, to asthma, to infectious diseases (Table 1). The cytotoxic effect of *Terminalia bentzoë*, on a panel of cancer cell lines, and its ability to impede the cell cycle progression in hepatocellular carcinoma (HepG2) cells were determined. The bioactive constituents in *T. bentzoë* leaf extract were characterized following bioassay-guided fractionation.

## 2. Results

### 2.1. Estimation of Polyphenols Level in the Investigated Leaf Extracts

The phenolic content varied significantly among the seven leaf extracts under study (*p* < 0.05), with amounts ranging between 70.2 ± 4.72 mg and 385 ± 24.1 mg gallic acid equivalent/g. The total flavonoid levels ranged between 2.43 ± 0.06 mg and 12.9 ± 0.45 mg quercetin equivalent/g. Based on the spectrophotometric assay results, the estimated levels of phenolics and flavonoids were significantly the highest (*p* < 0.05) in *T. bentzoë* leaf extract as compared to the other investigated leaf extracts (Table 2). While the proanthocyanidin content prevailed in the *E. sideroxyloides* leaf, both the *S. lineata* leaf and *T. bentzoë* leaf had a remarkably negligible amount of proanthocyanidin detected by the butanol/HCl assay.

### 2.2. In Vitro Antioxidant Activities of the Investigated Leaf Extracts

An array of five analytical models was used to benchmark the antioxidant potential of the investigated leaf extracts. The Mascarene endemic plant leaf extracts exhibited a varying degree of activities in the different antioxidant assays. All extracts showed a dose-dependent metal chelating and free radical scavenging activity (Table 2). In terms of iron chelation, all the extracts were weak chelators compared to EDTA, with an IC_50_ value of 0.01 ± 0.00 mg/mL (23.6 ± 0.22 µM) (*p* ≤ 0.05). As depicted in Table 2, among the seven accessions of plants, *T. bentzoë* showed the most effective antioxidant potential in all the five antioxidant assays. Thus, the free radical quenching activity of *T. bentzoë* was further evaluated in a hydroxyl radical scavenging assay. *T. bentzoë* (IC_50_ = 0.25 ± 0.03 mg/mL) exhibited a significantly (*p* < 0.0001) greater degree of protection against Fenton-mediated oxidative damage to 2-deoxyribose sugar moiety as compared to gallic acid (IC_50_ = 1.65 ± 0.09 mg/mL).

### 2.3. Effect of T. bentzoë Leaf Extract on Cell Survival

To assess the antiproliferative properties of the *T. bentzoë* leaf extract on cancer cells, first, the influence of the extracts on the cell viabilities of five cancer cell lines, notably human liposarcoma (SW872), human lung carcinoma (A549), human hepatocellular carcinoma (HepG2), and human ovarian carcinoma OVCAR-4 and OVCAR-8 cells, was investigated using the methyl thiazolyl diphenyl-tetrazolium bromide (MTT) assay. *T. bentzoë* suppressed the growth of all cancer cell lines in a dose-dependent manner. However, the dose of extract required to reach the half-maximal inhibitory concentration differed considerably among the various cancer cell types. The cytotoxicity of *T. bentzoë* was also evaluated against the non-malignant human ovarian epithelial (HOE) cells and the *IC_50_* value obtained (Table 3). HepG2 cells were more sensitive to *T. bentzoë* treatment (selective index value 2.4 compared to HOE cells) (Table 3). At 48 h of exposure, the highest concentration of *T. bentzoë* extract (100 µg/mL) reduced HepG2 cell viability to 20% as compared to the untreated control cells (Appendix A). Subsequently, the effect of *T. bentzoë* extract on the replicative ability of HepG2 cells was evaluated using the clonogenic cell survival assay. *T. bentzoë* treatment significantly (*p* < 0.0001) reduced the number of surviving HepG2 colonies as compared to untreated control cells (Figure 1).

### 2.4. Genotoxic Effect of T. bentzoë Extract in HepG2 Cells.

The DNA damage induced by the *T. bentzoë* leaf extract was assessed in HepG2 cells by the alkaline comet assay. Treatment with 10 µg/mL *T. bentzoë* leaf extract for 24 h resulted in the induction of significant (*p* ≤ 0.0001) DNA damage in HepG2 cells as compared to untreated control cells. The occurrence of DNA damaged was scored in terms of tail length, tail intensity and olive tail moment of cell cultures by comet assay software (Table 4**)**. The increased olive tail moment in response to *T. bentzoë* treatment was four-fold higher than that of untreated control cells. Cells treated with 200 µM H_2_O_2_ for 30 min were used as a positive control.

### 2.5. T. bentzoë Induced Cell Death

The proportion and distribution of HepG2 cells stained by annexin V- fluorescein isothiocyanate (FITC) and propidium iodide (PI) after 48 h of exposure to *T. bentzoë* extract are illustrated in Figure 2A and Appendix A. The results indicated that, at a concentration equivalent to 10 µg/mL and 20 µg/mL, the distribution pattern of the cell populations in the different quadrants (Appendix A) is similar to that of the dimethyl sulfoxide (DMSO) control. However, the exposure of HepG2 cells for 48 h at 40 µg/mL stimulated a significant increase in annexin V (1.41-fold, *p* < 0.01) and PI (1.52-fold, *p* < 0.001) fluorescence compared to the control. This increase is notably reflected in a higher percentage of necrotic cells (11.7%, *p* < 0.01) compared to the DMSO control (4.33%), indicating apoptotic/necrotic cell death at this test concentration. Etoposide significantly (*p* < 0.001) increased the proportion of cells undergoing apoptosis (between 10% and 12%)/necrosis (between 16% and 21%) at all three test doses, relative to DMSO control.

### 2.6. Cell Cycle Progression and T. bentzoë

The flow cytometric analysis of DNA content of the HepG2 cells treated with test extracts for 48 h allowed the determination of the percentage of the cell population in each phase of the cell cycle. As emphasized in Figure 2B and Appendix A, the exposure of HepG2 cells to 40 µg/mL of *T. bentzoë* induced G0/G1 cell cycle arrest by increasing the cell population in G0/G1 (until 73.1 ± 1.76%, *p* < 0.01 vs. ctrl) to the detriment of the G2/M phase (until 17.92 ± 1.76%, *p* < 0.001 vs. ctrl). In contrast, the treatment with 4 µg/mL etoposide led to the accumulation of G2/M cell fraction (until 53.6 ± 5.55%), to the detriment of the G0/G1 cells fraction (42.2 ± 6.25%), suggesting the G2/M arrest of HepG2 cell progression.

### 2.7. Bioassay Guided Fractionation of T. bentzoë Leaf Extract

#### 2.7.1. Effect of *T. bentzoë* Fractions on HepG2 Cell Viability

HepG2 cells were used as a model to fractionate and characterize the bioactive components present in *T. bentzoë* leaf extract. The first round of fractionation was achieved using liquid–liquid portioning with organic solvents of increasing polarities. The *T. bentzoë* butanol fraction, being the most active, was further fractionated on a Sephadex LH-20 column. A total of 11 subfractions was derived, with their cytotoxic activity being limited between fractions F4 to F10 only (Table 5). The assessment of their chromatographic patterns revealed differences in the chemical compositions of the fractions, albeit with multiple overlapping peaks highlighting the chemical complexity of the subfractions. Given the most potent cytotoxic activity of the subfraction F6 (IC_50_ = 15.2 ± 1.8 µg/mL), the further fractional separation of the latter was achieved using preparative high performance liquid chromatography (HPLC) to yield 11 HPLC subfractions (F6.1–F6.11). Of these, HPLC subfraction F6.1 retained most of the cytotoxic activity, providing an IC_50_ value of 15.8 ± 0.3 µg/mL. By contrast, HPLC fractions F2, F7, F9, F10 and F11 failed to effectively inhibit the growth of HepG2 cell cultures; hence no IC_50_ values were determined for these subfractions. A summary of the bioassay-guided fractionation employed is depicted in Figure 3.

#### 2.7.2. Antioxidant Potential of *T. bentzoë* Fractions

Both the ethyl acetate (22 ± 0.29 mmol Fe^2+^) and butanol (21.01 ± 0.56 mmol Fe^2+^) fractions were significantly (*p* ≤ 0.05) higher compared to the aqueous residual fraction. Furthermore, the FRAP value of the organic fractions was greater as compared to the total extract. The butanol fraction of *T. bentzoë* was a more effective scavenger of DPPH^•^ radical as opposed to the ethyl acetate and aqueous residual fractions (Table 5). As far as the butanol subfractions were concerned, notably F6, F7, and F8 had better antioxidant activities in the FRAP and superoxide radical scavenging assays (Table 5).

### 2.8. Characterization of the Cytotoxic Components of T. bentzoë

In an effort to identify the bioactive constituents in *T. bentzoë*, bioassay-guided fractionation was carried out. The liquid chromatography mass spectrometry (LC-MS) analysis, in conjunction with NMR spectroscopy, of the *T. bentzoë* butanol fraction 6 and its semi-prep HPLC sub-fractions allowed the identification of eight phenolic compounds, including α and β-punicalagin (**1**), isoterchebulin (**2**), terflavin A (**3**), 3,4,6-trigalloyl-β-D-glucopyranose (**4**), 2”-*O*-galloyl-orientin (**5**), 2”-*O*-galloyl-isoorientin (**6**), 2”-*O*-galloylvitexin (**7**), and ellagic acid (**8**) (Figure 4, Table 6, Appendix A). The bioactive HPLC subfraction F6.1 (IC_50_ value = 15.8 ± 0.3 µg/mL) was revealed to be α and β-punicalagin (**1**) (Appendix A), while a mixture of isoterchebulin (**2**), terflavin A (**3**) and 3,4,6-trigalloyl-β-d-glucopyranose (**4**) (Appendix A) was identified from the active subfraction F6.6 (IC_50_ value = 19.6 ± 0.7 µg/mL). A mixture of 2”-*O*-galloyl-orientin (**5**) and 2”-*O*-galloyl-isoorientin (**6**) (Appendix A), with a ratio of 5:2 based on the ^1^H NMR integration of single hydrogen in each compound, was identified from the non-active subfraction F6.9. 2”-*O*-galloylvitexin (**7**) and ellagic acid (**8**) were detected only by LC-MS from the butanol fraction 6 (Appendix A). Additionally, two simple phenolics, gallic acid and methyl gallate, were detected in both the ethyl acetate and butanol fractions, and identified by GC-MS analysis (Appendix A). They were likely present in other less bioactive fractions. The levels of gallic and methyl gallate were quantified as 15.6 ± 1.1 and 26.2 ± 8.1 µg/mg total extract (*n* = 3) by HPLC, respectively.

## 3. Discussion

Medicinal plants are known to be the epicenter of numerous well-established ethnomedicinal systems across the globe [4]. However, more than 84% of these medicinal plants have been poorly studied in regards to their phytochemical compositions, clinical efficacy, as well as their safety and toxicological profiles [4]. Moreover, forest cover is being uprooted across the world, at an unprecedented rate, threatening the survival of at least 15,000 medicinal plant species [30]. Plant secondary metabolites have contributed enormously to the modern-day pharmaceutical industry by providing the chemical backbone for almost 25% of the 1562 clinical agents, as well as 60% of the 246 oncologic drugs, approved by the US Federal Drugs Administration between 1981 and 2014 [1,2,31]. As such it is of utmost importance to evaluate the unexplored terrestrial flora for their therapeutic potential, as the dwindling medicinal plant species continue to stand as a rich repository to probe for novel chemotypes in the drug developmental process.

The current findings highlight the richness of the different subclasses of polyphenolics, notably phenolics, flavonoids and proanthocyanidins, the distribution of which differed significantly (*p* < 0.05) among the investigated accessions (Table 2). The biosynthesis of flavonoids, in particular flavonols, is known to be upregulated in response to ultra-violet radiation [32]. The accumulation of flavonoids in the endemic plant leaves collected from Mauritius may be attributed to the high sunlight conditions and UV radiation, which are characteristic to tropical islands like the Mascarene [33].

Given the ubiquitous involvement of oxidative damage in carcinogenesis, antioxidant-rich secondary metabolites, notably polyphenolics, have attracted much interest in the search of novel and alternative treatment modalities for cancer [34,35]. In this vein, in vitro antioxidant activities often correlated strongly with growth inhibitory activity against cancer cell lines [36,37]. However, it is crucial to note that both ROS and antioxidants have a “Janus-faced” effect in cancer management, as their effects differ in the early events of cancer initiation from those seen in the survival and propagation of established tumors [34,38]. On one hand, where a moderate ROS level promotes the transformation of normal cells to malignant cells, elevating the ROS level in cancer cells beyond the cell tolerance threshold may promote oxidative stress-induced cell cytotoxicity [38,39,40]. It should be noted that many standard chemotherapies cause cancer cell death via the generation of ROS and excessive oxidation [41]; therefore, the effect of antioxidant polyphenols in combination with these chemotherapies on cancers must be fully evaluated in vitro and in vivo before use in patients.

The antioxidant mechanisms of action of polyphenols are multifaceted, hence the panel of in vitro assay models used to gauge the antioxidant potential of the evaluated extracts [42]. *T. bentzoë*, having the highest abundance of total phenolics and flavonoids, also exhibited the most potent antioxidant activity in all six in vitro assays and was further evaluated for its cytotoxicity against cancer cell lines. However, the other six plant species might also show cytotoxicity through different mechanisms, which were not investigated in this study.

The cytotoxicity of different *Terminalia* species against multiple cancerous cell lines is documented. The investigation of the cytotoxic activity of methanolic leaf extracts of *T. arjuna* against human chronic myelogenous leukaemia cells led to the isolation of bioactive ursolic acid (triterpenoid) [43]. Leaf extracts of *T. chebula* suppressed the growth of human breast and lung cancer cell lines [44]. Along a similar line, the leaf extract from *T. catappa* significantly (*p* < 0.05) suppressed the proliferation of the human colorectal (SW480) cell line in a dose-dependent manner by downregulating the level of B-cell lymphoma 2 (BCL-2) gene expression while upregulating the levels of Caspase 9 and Caspase 3, indicative of the mitochondrial pathway of apoptosis in SW480 cells [45].

The foremost aim of oncologic agents is to precisely target cancerous cells with minimal effects on non-malignant cells. *T. bentzoë* crude extract had a selective index value of approximately 2.5, indicating that the extract had some degree of selectivity towards HepG2 cells as opposed to non-malignant human ovarian epithelial cells. Nevertheless, it is crucial to also assess the cytotoxicity of the extract against a panel of normal cells of different tissue origins, to provide greater insight regarding its toxicity window and safety profile.

In line with the United States National Cancer Institute cytotoxicity guidelines [46,47], *T. bentzoë* crude extract with an IC_50_ value of 22.8 ± 1.3 µg/mL can be considered as a potent candidate for further investigation with regards to its anticancer potential. It is known that, following exposure to toxicants, a subpopulation of cells enters a dormant state. These dormant viable cells further retain their ability to replicate into stem-like progeny cells, which might lead to the development of drug-resistance and cancer relapse [48,49]. Thus, while investigating the potential anticancer effect of extracts, it is warranted to also evaluate their long-term effect on the replicative ability of the cancer cell line. The colony formation assay is a simple and useful in vitro model, considered as the gold standard to predict the long-term sensitivity of cancer cell response to therapeutics treatment [48,50]. In this vein, the current study evidenced the ability of *T. bentzoë* to impede the replicative potential of HepG2 cells (Figure 1). Numerous anticancer agents, including polyphenols, are known to abrogate the limitless replicative ability of different cancer cell variants [51,52]. Catechin, catechin-3-*O*-gallate, 7-*O*-galloyl catechin and methyl gallate purified from *Acasis hydaspica* are reported to suppress the long-term clonal proliferation of prostate cancer (PC-3) cells [53].

This study also attempted to delineate the mode of *T. bentzoë*-induced HepG2 cell death. The flow cytometric analysis of annexin V-FITC and PI dual stained HepG2 cells treated with 40 µg/mL revealed a significant (*p* < 0.05) increase in both Annexin V-FITC and PI fluorescence levels, compared to the untreated control, thus indicating the putative activation of both apoptotic and necrotic pathways in HepG2 cells. Cells grown as monoculture are known to initiate apoptosis, which is terminated by necrosis-like events, also termed as secondary necrosis, due to the absence of phagocytic scavenger cells [54,55]. Numerous studies evidenced this type of cancer cell death following treatment with cytotoxic agents. Ellagic acid was reported to induce both apoptotic and necrotic cell death mechanisms in human pancreatic cancer cell cultures [56]. A similar effect of *Lepidium sativum* and *Vitis vinifera* extract was highlighted in human breast and skin cancer cells, respectively [57,58].

Plant extracts and/or the thereby-derived phytochemicals are known to provoke cancer cell death by causing oxidative damage to genetic material-mediated cell cycle arrest and subsequent cell death [59,60]. With this in mind, the DNA damage to HepG2 cellular DNA following *T. bentzoë* crude extract treatment was investigated using the alkaline comet assay, and scored in terms of tail length, tail intensity and olive tail moment. The comet assay is a well-established and highly sensitive method for the detection of DNA damage and fragmentation pattern [61]. *T. bentzoë* induced 4-fold significantly (*p* < 0.001) higher DNA damaged to HepG2 cells compared to the untreated control, as reflected by the olive tail moments (Table 4). Consistent with the DNA damaging ability, *T. bentzoë* halted the cell cycle progression significantly (*p* < 0.01, versus control) in the G0/G1 phase (Figure 2). A similar observation was reported in lung cancer cells, where casuarinin, a tannin purified from *T. arjuna* L. bark, provoked the apoptotic mechanism via DNA fragmentation and G0/G1 phase cell cycle arrest [62].

It is noteworthy that the cytotoxicity of the extract may arise from the complex interplay of the cocktails of secondary metabolites present, which may act either synergistically or antagonistically to produce the overall results. Furthermore, crude extracts often comprised a pool of inactive phytoconstituents that dilute the efficacy of the active components [63,64]. It is therefore of paramount importance to purify and identify the principal bioactive components to conduct further mechanistic studies to establish their molecular mode of action [63,65]. Moreover, isolating the lead compounds also allows for potential structural modification in an attempt to enhance their selectivity and potency [66]. As such, the MTT-guided fractionation revealed that only 6 out of the 11 preparative HPLC subfractions retained the potent cytotoxicity of the crude extract (Figure 3). The HPLC subfraction F6.1 was 1.4-fold more potent as compared to the crude extract.

LC-MS analysis in conjunction with NMR spectroscopy allowed the identification of eight phenolic compounds from the most bioactive subfraction F6. Punicalagin, isoterchebulin and ellagic acid have been previously reported from the bark extract of the same species collected in Réunion Island [24]. Although the other identified compounds are being reported for the first time, to the best of our current knowledge, in *T. bentzoë* leaf, some were reported in the leaf extracts of other *Terminalia* species. For instance, 2”-*O*-galloylvitexin and gallic acid were identified from *T. brachystemma* Welw. ex Hiern and *T. mollis* M. Laws leaves, respectively [26]. Likewise, terflavin A was found in the *T. catappa* L. leaf [28].

Punicalagin and ellagic acid were reported to induce S Phase arrest and G0/G1 phase arrest in HepG2 cells, respectively [67]. Moreover, punicalagin induced G0/G1 phase arrest in papillary human thyroid carcinoma (BCPAP) cells via the NF-κB signaling pathway [68]. In human ovarian cancer (A2780) cells, punicalagin treatment was associated with an increase in the number of cells arrested at the G1/S phase, as well as the downregulation of the β-catenin signaling pathway [69]. Ellagic acid administration in prostate cancer patients was associated with decreased prostate-specific antigen, as well as reduced chemotherapy-induced myelotoxicity [70]. It may be confidently proposed that the overall cytotoxicity of *T. bentzoë* leaf extract may be a synergistic effect of the identified compounds. Gallic acid was reported to induce S phase arrest in HepG2 cells [71] and be cytotoxic to ovarian cancer cells [72]. Furthermore, gallic acid was also shown to impair centrosomal clustering in human cervical cancer (Hela) cells, thus causing a mitotic catastrophe and ultimate cell G2/M phase cell cycle arrest.

## 4. Materials and Methods

### 4.1. Plant Material and Preparation of Total Extracts

The healthy fresh leaves of seven Mascarene endemic plants were collected in Mauritius and deposited at the Mauritius herbarium, where plant species were authenticated by Kersley Pynee, National Parks & Conservation Service, Mauritius (Table 1). The leaves were air-dried followed by exhaustive maceration with aqueous methanol (80%, *v/v*) and freeze-dried as described previously [73]. The assay results were expressed in terms of the lyophilized weight of extracts.

### 4.2. Estimation of Polyphenolic Contents

The total phenolic, flavonoid, and proanthocyanidin level in the crude extracts were estimated using the Folin–Ciocalteu assay, aluminium chloride assay and HCl/Butan-1-ol assay as described [73].

### 4.3. In Vitro Antioxidant Capacities of Extracts

The antioxidant potential of the extracts was investigated according to reported methods [73,74]. Extract vehicle and gallic acid (or otherwise stated) were used as negative and positive controls, respectively. The percentage activity of the extracts was calculated relative to the negative control. GraphPad Prism 6.01 software (GraphPad, Inc., San Diego, CA, USA) was used to plot the dose–response curves and to generate the half-maximal inhibitory concentration (IC_50_) values. All experiments were performed in triplicates in three independent assays. The results were expressed as mean ± SEM.

#### 4.3.1. Ferric Reducing Antioxidant Potential (FRAP) Assay

The final reaction volume of 3.4 mL of the ferric reducing antioxidant power assay contained 100 µL of extract and 300 µL of water followed by the addition of 3 mL FRAP reagent. The FRAP reagent was prepared immediately before use by mixing 100 mL of 0.25 M acetate buffer (pH 3.6), 10 mL of 20 mM ferric chloride (source of Fe^3+^) and 10 mL of 10 mM 2,4,6-tripyridyl-s-triazine. After incubating the mixture for 4 min at ambient temperature, the absorbance was read at 593 nm against a blank. Results were reported as in mmol Fe^2+^.

#### 4.3.2. 2,2-Diphenyl-1-picrylhydrazyl (DPPH) Assay

The DPPH assay protocol involved mixing varying concentrations, between 0 and 25 µg/mL, of 100 µL of the methanolic extract with 200 µL of 100 µM methanolic DPPH and absorbances were read at 492 nm, 30 min post-incubation at ambient temperature.

#### 4.3.3. Iron Chelating Assay

The reaction mixture for the iron chelating activity contained 40 µL of plant extract (concentrations between 0 and 10 mg/mL), 10 µL of FeCl_2_⋅4H_2_O (0.5 mM) and 150 µL of distilled deionized water. The mixture was incubated at ambient temperature for 5 min, before the addition of 10 µL of ferrozine (2.5 mM) and the absorbance was read at 562 nm.

#### 4.3.4. Superoxide Scavenging Assay

The final 250 µL reaction volume for the superoxide anion scavenging assay contained 25 µL of plant extract (concentrations between 0 and 300 µg/mL), 100 µL of 156 µM of nitroblue tetrazolium, 100 µL of 200 µM beta-nicotinamide adenine dinucleotide reduced disodium salt hydrate and 30 µL of phenazine methosulphate. The absorbance was read at 560 nm following 30 min incubation at 25 °C.

#### 4.3.5. Nitric Oxide Scavenging Assay

The nitric oxide radical scavenging activity was conducted in a 96-well plate. In total, 50 µL of aqueous extract (0 to 100 µg/mL) and 100 µL of 5 mM of sodium nitroprusside (in phosphate saline buffer, pH 7.4) was incubated at 25 °C for 150 min. After incubation, 125 µL of the reaction mixture was transferred to another 96-well plate, to which 100 µL of 0.33% sulfanilic acid in 20% glacial acetic acid was added. After 5 min, 100 µL of 0.1% of *N*-1-napthyethylenediamine dihydrochloride was added and the pink coloration formed was read at 546 nm.

#### 4.3.6. Deoxyribose Degradation Inhibitory Assay

The deoxyribose degradation inhibitory assay protocol was optimized to a 24-well microtiter plate format. Each well contained 50 µL of aqueous extract, 50 µL of 1 mM EDTA, 100 µL of 500 µM FeCl_3_, 50 µL of 1 mM ascorbic acid, 50 µL of 10 mM hydrogen peroxide, 100 µL of 100 mM KH_2_PO_4_-KOH buffer (pH 7.4) and 100 µL of 15 mM 2-deoxyribose. The reaction mixture was incubated at 37 °C for 90 min. At the end of the incubation period, 500 µL of 10% (*w/v*) trichloroacetic acid followed by 500 µL of 1% (*w/v*) thiobarbituric acid were added to each well and the solutions were heated in a water bath at 80 °C for 20 min to develop the pink chromogen. The absorbance of the reaction mixture was read at 532 nm, both before and after incubation. Results were given in mg lyophilized extract/mL.

### 4.4. Human Cell Lines and Culture Conditions

Human liposarcoma cells (SW872), human lung carcinoma cells (A549), human hepatocellular carcinoma cells (HepG2), and human ovarian carcinoma cell lines OVCAR-4 and OVCAR-8 were purchased from American Type Culture Collection (USA). Human ovarian epithelial (HOE) cells immortalized using SV40 large T antigen were obtained from Applied Biological Materials Inc (Richmond, BC, Canada). All cell lines, except the ovarian cell lines, were cultured in Dulbecco’s Modified Eagle’s Medium. Roswell Park Memorial Institute (RPMI) 1640 medium was used in the case of ovarian cell lines. Culture medium was supplemented with 10% fetal bovine serum, 2 mM L-glutamine and 100 U/L streptomycin-penicillin. Cells were grown in a humidified atmosphere of 5% carbon dioxide and 95% humidity at 37 °C.

### 4.5. Cell-Based Assays

#### 4.5.1. MTT Viability Assay

The viability of the investigated cells treated with test samples was evaluated using the MTT cell viability assay as previously described [75]. Following the overnight acclimatization of cells in 96-well plate, cells were treated with test samples for 48 h and assayed for different parameters. For the 96-well plate, the seeding densities for cancer cell lines and HOE cells were 5 × 10^3^ cells and 2 × 10^3^ cells per well, respectively. All experiments were performed in triplicates (unless otherwise specified) in three independent assays. To compare the chemotherapeutic potential of the active extracts, the cytotoxicity of the extracts/purified compounds was evaluated against etoposide, a clinically used oncologic agent. The percentage cell viability relative to DMSO control (0.025% *v/v*, final concentration) was calculated and the IC_50_ value determined using GraphPad Prism 6.01 software (GraphPad, Inc., San Diego, CA, USA).

#### 4.5.2. Clonogenic Cell Survival Assay

The effect of the *T. bentzoë* leaf extract on the cell reproductive death was assessed by clonogenic cell survival assay according to reported methods, with slight modifications [76,77]. HepG2 cells were seeded in a 6-well plate (500/well) and allowed to attach overnight. Following a 48-h treatment period with plant extract/control, the media was replaced with fresh complete culture media and cells were grown under standard recommended culture conditions for an additional 7 days to allow large colonies’ formation. Colonies were then fixed with 4% paraformaldehyde for 30 min and stained with 0.5% (*w/v*) crystal violet. The individual wells were imaged using a digital camera and the colonies counted using ImageJ software, version 1.51 (the US, National Institute of Health). The cytotoxic effect was expressed as the percentage of surviving colonies relative to the untreated control.

#### 4.5.3. Single Cell Gel Electrophoresis

Comet assay was carried out according to the method described [78,79] with minor modifications. Briefly, 30 µL of pre-treated cells were mixed with 70 µL of 1% (*w/v*) low melting agarose (LMA) and 40 µL of the cell–LMA mixture was placed on frosted microscope slides pre-coated with 1.5% normal melting agarose. A coverslip was placed on top of the cell–LMA mix and allowed to solidify at 4 °C for 1 h in the dark. Following solidification, the coverslip was gently slid off and the slides were immersed in pre-chilled lysis buffer (2.5 M NaCl, 0.1 M EDTA, 10 mM Tris base, 1% *v/v* Triton X-100 (added 30 min before use), pH 10, 4 °C) for 1 h in the dark. Following this period, the excess lysis solution was drained and the slides were submerged in electrophoresis buffer (0.2 M NaOH, 1 mM EDTA, pH 13, 4 °C) for 30 min in the dark, to allow the DNA to unwind.

Electrophoresis was conducted for 30 min at 30 volts and 350 mA. The gels were neutralized by immersing in pre-chilled neutralization buffer (0.4 Tris-HCL, pH 7.5, 4 °C) for 10 min in the dark. The slides were washed in distilled water, fixed with 4 % formalin solution for 20 min and allowed to air-dry overnight. The slides were stained with Hoechst 33342 (1 µg/mL), air-dried in the dark and visualized at 200× magnification in DAPI light cube (Ex: 357/44 nm; Em: 447/60 nm), using an EVOS fluorescence microscope (Life Technologies). Damaged DNA was measured for 100 randomly selected cells (for each independent experiment) using the Comet Assay IV 4.3.1 (Perceptive instrument, Suffolk, UK).

#### 4.5.4. Flow Cytometric Analysis

Apoptosis/necrosis analysis was performed on HepG2 cells after 48 hours’ treatment by flow cytometry (Beckman Coulter’s CytoFLEX and Cytexpert software, version 2.4) using Annexin V-FITC and propidium iodide (PI) double staining as described in a previous study [80]. Cell cycle analysis was performed using propidium iodide for DNA staining as described [81]. The percentages of cells in different phases (G0/G1, S and G2M phases) were quantified from propidium iodide fluorescence intensity-area (PI-A) histograms corresponding to the DNA content of HepG2 cells.

#### 4.5.5. MTT-Guided Fractionation and Identification of Bioactive Molecules

The total extract of *T. bentzoë* was solubilized in distilled water and sequentially partitioned with ethyl acetate, followed by n-butanol. Each fraction was dried and its cytotoxicity evaluated against SW872, A549 and HepG2 using the MTT cell viability assay. The butanol fraction selective towards HepG2 cells was subjected to Sephadex LH-20 column chromatography (30 × 2.1 cm internal diameter) and eluted with water, water/methanol (3:1 *v*/*v*, 1:1 *v*/*v* and 1:3 *v*/*v*, respectively), methanol and acetone. The flow rate was maintained at 1.5 mL/min. Guided by the cytotoxicity against HepG2 cells and purity profile, potent sub-fractions were further fractionated using a semi-preparative HPLC column. The crude extract and thereof-derived purified sub-fractions were analyzed by extensive spectroscopic methods including GC-MS, LC-MS, HPLC, ^1^H-NMR and ^13^C-NMR (Supporting Information). For analytical HPLC, the concentration of standards in the crude extract was determined from the linear regression of the analytical standard curves, namely *y* = 12.317 *x*, *R*^2^ = 0.9996: gallic acid; and *y* = 16.066 *x*, *R*^2^ = 0.9992: methyl gallate.

### 4.6. Statistical Analysis

Statistical analyses were performed using GraphPad Prism 6.01 software (GraphPad Inc., San Diego, CA, USA). The mean values among extracts were compared using One-Way ANOVA. Student *t*-test and/or Tukey’s multiple comparisons as Post Hoc test were used to determine significances in mean phytochemicals, antioxidants and cytotoxic activities among different species. All charts were generated using GraphPad Prism 6.01 software (GraphPad Inc., San Diego, CA, USA).

## 5. Conclusions

The findings evidenced the selective long-term cytotoxicity of the antioxidant-rich *T. bentzoë* leaf extract against HepG2 cells. This plant, which is also used traditionally in the mitigation of asthma, hemorrhages, diarrhea, and sexually transmissible diseases, amongst others, has also shown some in vitro anticancer activities against HepG2 cells. The cytotoxic nature of *T. bentzoë* leaf extract against cancerous cells indicated that the *T. bentzoë* leaf has the potential to be repurposed in the mitigation of cancer as part of traditional medicine. The results so far generated support the hypothesis that *T. bentzoë* extract induces apoptosis/necrosis cell death in HepG2 cells via the degradation of cellular genetic material and the subsequent arrest of the cell cycle’s progression in the G0/G1 phase. Overall, the MTT-guided fractionation of the crude extract and spectroscopic analyses allowed the characterization of 10 phenolic compounds, including punicalagin, ellagic acid, gallic acid and methyl gallate, that are known to have established *in vitro* and *in vivo* anticancer properties. However, the contribution of the other non-identified phytoconstituents cannot be excluded, and thus further evaluation of the identified components is warranted alongside the crude extract. The metabolite profiling of the leaf extract may be envizaged in future studies. Taking into consideration the guidelines of the US national cancer institute as well as the selectivity index observed in this study, *T. bentzoë* leaf extract has been revealed as a promising candidate that can be exploited in the search for novel anticancer agents. The current study provided base line data for more in-depth investigations of the chemotherapeutic potential of *T. bentzoë* plant extracts. Further mechanistic investigation, directed towards delineating the molecular mechanisms via which the purified bioactive entities target the aberrant signaling pathways involved in carcinogenesis, is needed to fuel *in vivo* studies. Future animal studies and ideally human clinical trials are warranted to establish the physiological doses for human administration.

## Figures and Tables

**Figure 1 pharmaceuticals-13-00303-f001:**
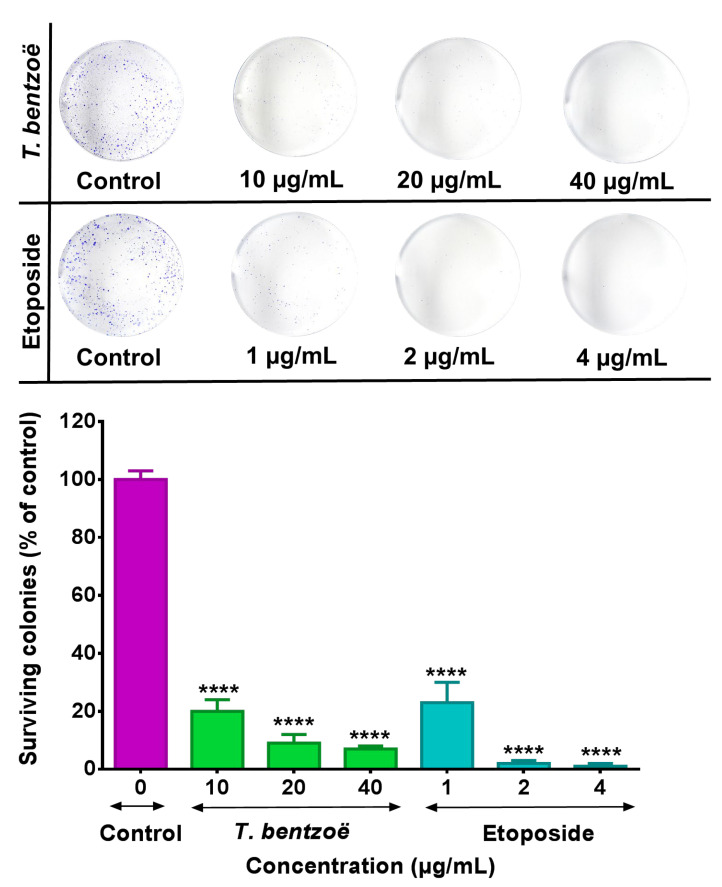
HepG2 cells were treated with the indicated concentrations (µg/mL) of test extracts for 48 h and subsequently allowed to grow into colonies for 14 days. After 14 days, the colonies were stained with 0.1% crystal violet and the images of the wells were captured using a digital camera. The colonies were counted using Image J software. Each experiment was performed three times. Asterisks represent significant differences between extract treatments and untreated control, **** *p* ≤ 0.0001.

**Figure 2 pharmaceuticals-13-00303-f002:**
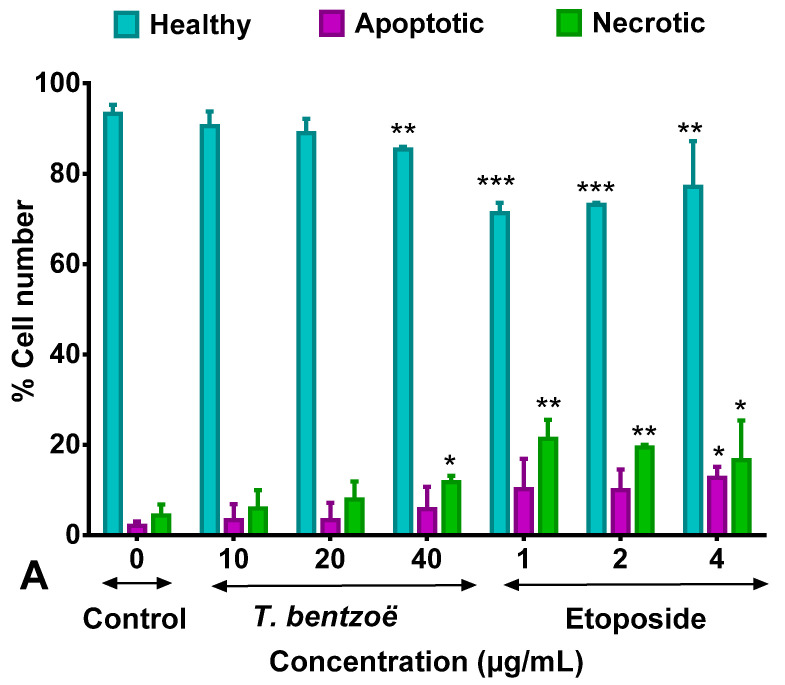
Effect of *Terminalia bentzoë* and etoposide on HepG2 cells as analyzed by flow cytometry. (**A**) Annexin V-FITC/PI staining of HepG2 cells after 48 hours’ treatment and (**B**) cell cycle progression. Apoptosis and necrosis levels are expressed as percentage of the total cell population and given as mean ± SD (*n* = 5). Percentage of cells in different phases (G0/G1, S and G2/M phases) are expressed as mean ± SD (*n* = 3). Asterisks represent significant differences between test concentrations and dimethyl sulfoxide (DMSO) control. * *p* ≤ 0.05, ** *p* ≤ 0.01, *** *p* ≤ 0.001, **** *p* ≤ 0.0001.

**Figure 3 pharmaceuticals-13-00303-f003:**
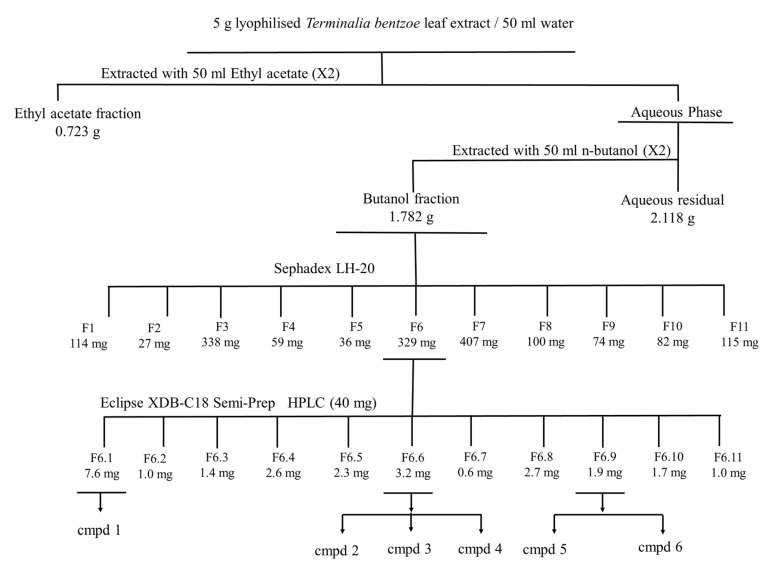
MTT-guided fractionation of *T. bentzoë* extract cytotoxicity against HepG2 cells. Cmpd: Compound. Cmpd 1= α and β- Punicalagin; compd 2 = Isoterchebulin; compd 3 = Terflavin A; cmpd 4 = 3,4,6-trigalloyl-β-D-glucopyranose; cmpd 5 = 2”-*O*-galloyl-orientin; cmpd 6 = 2”-*O*-galloyl-isoorientin.

**Figure 4 pharmaceuticals-13-00303-f004:**
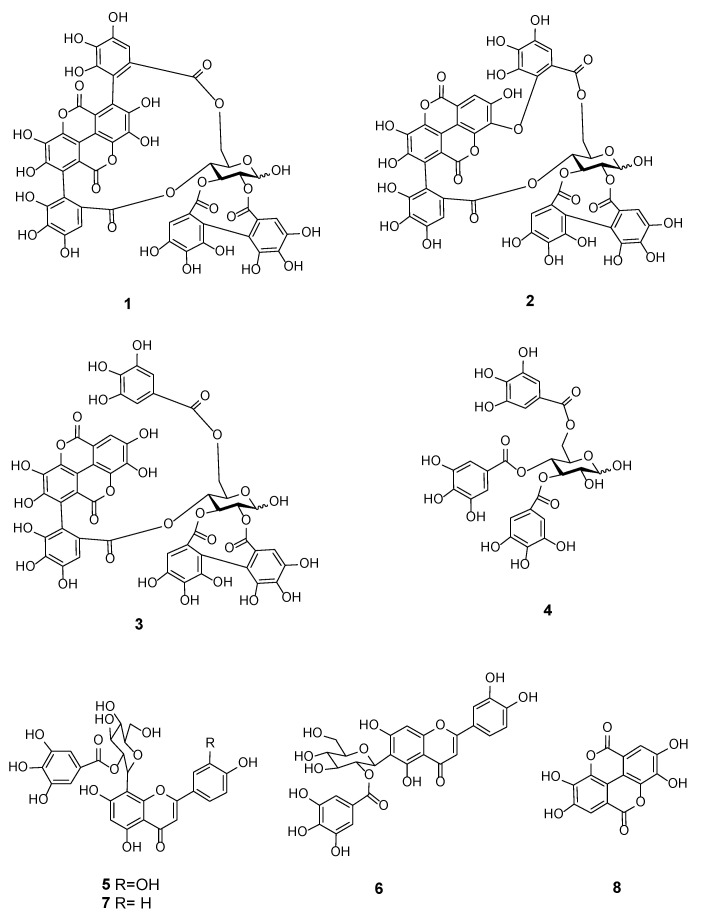
Chemical structure of polyphenolic compounds (**1**–**8**) identified in *Terminalia bentzoë* L. leaf extract.

**Table 1 pharmaceuticals-13-00303-t001:** The investigated Mascarene endemic plant species.

Species	Family	Vernacular Names	Ethnomedicinal Uses [16]	Collection Site	Collection Date	Mauritius Herbarium Accession Code	% Yield
*Antirhea borbonica* J.F.Gmel	Rubiaceae	Bois lousteau, Bois d’oiseau	Astringent, diarrhea, dysentery, stop bleeding, promote wound repair, skin diseases, tambave, Urinary tract infections	Gaulettes Serrées	14 October 2014	MAU 0009462	5.53
*Dictyosperma album* (Bory) H. Wendl. & Drude ex Scheff var. *conjugatum* H. E. Moore & Guého	Arecaceae	Palmiste blanc	Not described	Réduit, Joseph Guého Arboretum	19 August 2014	MAU 0016674	8.52
*Erythroxylum sideroxyloides* Lam	Erythroxylaceae	Bois de ronde	Renal stones	Lower Gorges National Park, ‘Morne Sec’	15 October 2014	MAU 0016542	13.81
*Ficus mauritiana* Lam	Moraceae	Figuier du pays	Not described	Gaulettes Serrées	14 November 2014	MAU 0011002	3.10
*Hancea integrifolia* (Willd.) S.E.C. Sierra, Kulju and Welzen	Euphorbiaceae	Bois pigeon	Clean the blood and improve blood circulation, tonic.	Gaulettes Serrées	14 November 2014	MAU 0016431	10.42
*Stillingia lineata* Muell. Arg	Euphorbiaceae	Fangame; Bois de lait; Tanguin de pays	Eczema, skin disease	Lower Gorges National Park, ‘Morne Sec’	27 November 2014	MAU 0016545	6.28
*Terminalia bentzoë* (L.) L.f. subsp. *bentzoë*	Combretaceae	Bois benjoin	Asthma, antipyretic, antimalarial, chills, dysentery, diarrhoea, depurative, emmenagogue, haemorrhages, Sexually transmissible diseases	Réduit, Joseph Guého Arboretum	7 October 2014	MAU 0016557	7.29

**Table 2 pharmaceuticals-13-00303-t002:** Phenolic content and antioxidant potential of investigated leaf extracts.

Extract	Total Phenolics ^1^	Total Flavonoids ^2^	Total Proanthocya Nidins ^3^	FRAP ^4^	Iron Chelating Activity ^5^	DPPH Scavenging Activity ^6^	Superoxide Scavenging Activity ^6^	Nitric Oxide Scavenging Activity ^6^
*A. borbonica*	70.2 ± 4.72 ^e^	3.15 ± 0.07 ^d,e^	5.71 ± 0.09 ^e^	3.32 ± 0.16 ^d,e,^****	4.05 ± 0.26 ^b,c,^****	11.2 ± 1.63 ^d,^****	19.0 ± 2.46 ^d,^****	80.3 ± 29.0 ^b,^**
*D. album*	75.7 ± 5.22 ^d^	2.43 ± 0.06 ^f^	30.9 ± 0.58 ^c^	2.21 ± 0.05 ^e,^****	4.83 ± 1.49 ^c,d,^****	7.89 ± 0.13 ^c,^****	32.7 ± 1.16 ^f,^****	68.0 ± 10.0 ^a,^*
*E. sideroxyloides*	182 ± 10.5 ^b^	3.62 ± 0.13 ^d^	121 ± 3.25 ^a^	9.14 ± 0.85 ^b,^****	1.46 ± 0.02 ^a,^****	4.44 ± 0.26 ^b,^****	12.7 ± 0.65 ^c,^****	24.3 ± 1.39 ^a^
*F. mauritiana*	133 ± 2.36 ^c^	10.4 ± 0.21 ^b^	85.7 ± 2.38 ^b^	5.38 ± 0.01 ^c,^****	0.43 ± 0.01 ^a,^****	5.35 ± 0.23 ^b,^****	24.2 ± 0.44 ^e,^****	87.08 ± 28.9 ^b,^***
*H. integrifolia*	142 ± 4.91 ^c^	2.75 ± 0.06 ^e,f^	18.4 ± 0.78 ^d^	9.37 ± 0.29 ^b,^****	0.78 ± 0.01 ^a,^****	4.16 ± 0.16 ^b,^****	9.55 ± 0.78 ^b,^***	68.49 ± 37.5 ^a,b,^**
*S. lineata*	97.7 ± 3.36 ^d^	6.61 ± 0.19 ^c^	ND	4.53 ± 0.02 ^c,d,^****	6.45 ± 0.02 ^d,^*	4.04 ± 0.17 ^a,b,^****	3.81 ± 0.48 ^a^	68.5 ± 42.9 ^b,^**
*T. bentzoë*	385 ± 24.1 ^a^	12.9 ± 0.45 ^a^	ND	18.2 ± 0.01 ^a,^****	0.10 ± 0.00 ^a,^****	2.65 ± 0.14 ^a,^***	5.20 ± 0.53 ^a^	9.74 ± 3.13 ^a^
Gallic acid	-	-	-	24.8 ± 0.22	8.00 ± 0.04 (47.0 ± 0.23 mM)	0.62 ± 0.05 (4.18 ± 0.32 µM)	5.52 ± 0.11 (31.4 ± 0.84 µM)	9.61 ± 1.75 (68.0 ± 13.9 µM)

^1^ Values are expressed as mg of gallic acid equivalent (GAE)/g; ^2^ values are expressed as mg of quercetin equivalent (QE)/g; ^3^ values are expressed as mg of cyanidin chloride equivalent (CCE)/g; ^4^ values are expressed in mmol Fe^2+^; ^5^ IC_50_ values are expressed in mg/mL; ^6^ IC_50_ values are expressed in µg/mL; Data represent mean ± standard error of mean (*n* = 3). ND = Not detected. Different letters between rows in each column represent significant differences between extracts (*p* < 0.05). Letter “a” represent the most potent/ highest abundance, while letter “f” least potent/lowest abundance in terms of antioxidant activity and polyphenolics content, respectively. Asterisks represent significant differences between extracts and gallic acid (positive control), * *p* ≤ 0.05, ** *p* ≤ 0.01, *** *p* ≤ 0.001, **** *p* ≤ 0.0001.

**Table 3 pharmaceuticals-13-00303-t003:** Cytotoxicity (IC_50_ µg/mL) of *T. bentzoë* against human cancer cell lines.

Sample	SW872	A549	HepG2	OVCAR-4	OVCAR-8	HOE
*T. bentzoë* extracts	45.4 ± 1.8 ****	96.8 ± 4.9 ****	22.8 ± 1.3 ****	30.1 ± 2.3	38.5 ± 4.2	55.5 ± 9.1
Etoposide	2.5 ± 0.2	6.8 ± 0.7	1.7 ± 0.2	NA	NA	1.5 ± 0.1

Data represent mean calculated IC_50_ values with a standard error of the mean (*n* = 3). NA = Etoposide at 1 µg/mL. Inhibited above 80% cancer cell growth, indicating a much lower concentration is required for determination of IC_50_ value. Asterisks represent significant differences between *T. bentzoë* and etoposide (positive control), **** *p* ≤ 0.0001.

**Table 4 pharmaceuticals-13-00303-t004:** Tail length, tail intensity and olive tail moment of H_2_O_2_ and *T. bentzoë* treated HepG2 cells.

Extracts and Controls	Tail Length (µm)	Tail Intensity	Olive Tail Moment
Negative control (Culture medium)	32.8 ± 0.3	0.8 ± 0.1	0.2 ± 0.0
*T. bentzoë* extract 10 µg/mL)	38.5 ± 0.4 ****	3.7 ± 0.2 ****	0.8 ± 0.0 ****
Positive control (200 µM H_2_O_2_)	90.8 ± 7.2 ****	46.8 ± 2.6 ****	14.1 ± 0.6 ****

Asterisks represent significant differences between plant extract and untreated control, **** *p* ≤ 0.0001.

**Table 5 pharmaceuticals-13-00303-t005:** Cytotoxic and antioxidant potential of *T. bentzoë* leaf fractions and the isolated punicalagin (1).

*T. bentzoë* Fractions		IC_50_ (µg/mL) Against HepG2 Cells	FRAP ^1^	DPPH ^2^	Superoxide Scavenging Activity ^2^
Ethyl Acetate		20.8 ± 0.1	21.98 ± 0.29 ^a,b^	1.17 ± 0.02 ^d,e^	7.43 ± 0.14 ^b^
Butanol		18.3 ± 3.4	21.01 ± 0.56 ^b,c^	0.98 ± 0.06 ^e^	7.65 ± 0.10 ^b^
Aqueous Residual		26.8 ± 5.5	15.29 ± 0.16 ^e^	1.78 ± 0.07 ^b,c^	10.70 ± 0.20 ^a,b^
*T.bnetzoe butanol* Subfractions	F1	ND	3.04 ± 0.10 ^g^	ND	ND
	F2	ND	13.62 ± 0.16 ^f^	2.65 ± 0.04 ^a^	16.90 ± 0.35 ^a^
	F3	ND	13.80 ± 0.14 ^e,f^	1.79 ± 0.12 ^b,c^	10.80 ± 0.28 ^a,b^
	F4	28.9 ± 1.5	14.42 ± 0.27 ^e,f^	2.09 ± 0.08 ^b^	8.96 ± 0.12 ^b^
	F5	25.7 ± 2.9	18.46 ± 0.15 ^d^	1.71 ± 0.07 ^c^	8.71 ± 0.10 ^b^
	F6	15.7 ± 1.8	23.01 ± 0.68 ^a^	1.19 ± 0.09 ^d,e^	7.03 ± 0.20 ^b^
	F7	19.9 ± 4.7	20.94 ± 0.49 ^b,c^	1.01 ± 0.07 ^d,e^	7.15 ± 0.18 ^b^
	F8	24.6 ± 5.9	19.86 ± 0.29 ^c,d^	1.12 ± 0.06 ^d,e^	8.76 ± 0.08 ^b^
	F9	22.3 ± 3.1	18.82 ± 0.3 ^d^	0.99 ± 0.06 ^e^	9.33 ± 0.24 ^b^
	F10	26.7 ± 6.2	18.47 ± 0.28 ^d^	1.34 ± 0.04 ^d^	9.05 ± 0.23 ^b^
	F11	ND	0.09 ± 0.02 ^h^	ND	ND
Active HPLC Subfractions	F6.1 (Punicalagin (**1**))	15.8 ± 0.3	-	-	-
	F6.3	27.5 ± 0.8	-	-	-
	F6.4	17.4 ± 0.4	-	-	-
	F6.5	22.1 ± 0.6	-	-	-
	F6.6	19.6 ± 0.7	-	-	-
	F6.8	18.6 ± 0.4	-	-	-

^1^ Values are expressed in units of mmol Fe^2+^/gFDW; ^2^ Values are expressed in units of µg/mL; Data represent mean with a standard error of the mean (*n* = 3). Different letters between rows in each column represent significant differences between extracts (*p* < 0.05). Letter “a” represent the most potent antioxidant activity, while letter “g” represent the least potent activity. ND = IC_50_ value not reached at maximum test dose. “-“ = Not tested.

**Table 6 pharmaceuticals-13-00303-t006:** Identification of phenolic compound in *T. bentzoë* butanol fraction F6 using high-resolution mass spectrometry, NMR spectroscopy, and comparison with the literature and available data.

Compound Number	RT/min	Negative ESI-MS [M − H]^−^	Molecular Formula	Compound	References
**1**	0.76	1083.0555	C_48_H_28_O_30_	Punicalagin	[24,25,26,27]
**2**	4.08	1083.0555	C_48_H_28_O_30_	Isoterchebulin	[24]
**3**	4.58	1085.0710	C_48_H_30_O_30_	Terflavin A	[28]
**4**	4.97	635.0867	C_27_H_24_O_18_	3,4,6-trigalloyl-β-d-glucopyranose	[29]
**5**	6.25	599.1017	C_28_H_24_O_15_	2”-*O*-galloyl-orientin	[25]
**6**	6.25	599.1017	C_28_H_24_O_15_	2”-*O*-galloyl-isoorientin	[25]
**7**	6.53	583.1074	C_28_H_24_O_14_	2”-*O*-Galloylvitexin	[25,26]
**8**	6.72	300.9987	C_14_H_6_O_8_	Ellagic acid	[24]

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
