# Peer review of "Terminalia bentzoë, a Mascarene Endemic Plant, Inhibits Human Hepatocellular Carcinoma Cells Growth In Vitro via G0/G1 Phase Cell Cycle Arrest"

_pharmaceuticals, 2020, doi:10.3390/ph13100303_

Round 1
Reviewer 1 Report
The MS describes anti-oxidative and anti- cancer activities of several leaf extracts of Mascarene endemic plants. One of the plants, Terminalia bentzoë, which had the highest activity, was further analyzed for activity and composition of extract fractions.
The MS seems to be premature for publication in Pharmaceuticals, since it has a relatively low level of novelty. This is because of several reasons.
- Most of the identified compounds in Terminalia bentzoë are known as well as their activity. Moreover, these compounds were identified in the present study from wild grown plant. Hence, it might very well be that composition of compounds may vary between leaves, branches, between trees, between growth conditions. Domestication or at least analysis of leaves from different tree branches or growth conditions should be done.
- What is the rational for examining anti-oxidative and anti-cancer activities? Evidently, part of the anti-cancer activity is oxidation of cancer cells. High ROS levels can become detrimental and induce cell death and numerous standard chemotherapies are cytotoxic towards cancer cells owing to their ability to induce drastic increases in ROS levels (Gorrini, C., Harris, I.S. and Mak, T.W., 2013. Modulation of oxidative stress as an anticancer strategy. Nature reviews Drug discovery, 12(12), pp.931-947).
- The authors did not prove that the identified compounds in Terminalia bentzoë active fractions are indeed the active ones. It is not enough to show the composition of active fraction but rather, activity of purified compounds and their composition should be determined.
- The apoptotic and cell cycle activities determined for Terminalia bentzoë are relatively low. No major changes in cell cycle are evidence.
- Concentrations of extract being used are relatively high, and may indicate low specificity. On top of that, the activity on normal cells, i.e., HOE, is quite similar to that on the cancer cell lines (Table 3 is missing HOE data).
As such, this study does not serve as a promising basis for further in-depth investigations on the potential use of T. bentzoë as supportive therapy in cancer management, as suggested. Rather, further study is needed to solidify the findings and promote novelty.
Author Response
Thank you. Please see the attached document.

Reviewer 2 Report
Rummun et al., describe “Focus on Mascarenes endemic plants with specific phytochemical composition, potent antioxidant and antiproliferative properties”. The result shown that T. bentzoë extract suppressed the growth of human hepatocellular carcinoma cells and significantly halted the cell cycle progression in G0/G1 phase, decreased the cells replicative potential and induced significant DNA damage.
Comments:
- Please clarify the abstract. The objectives should be clearly addressed.
- Include a better rationalization for choice of Mascarene endemic plants leaves.
- Since there are several mechanisms are involved in G0/G1 phase, it is highly recommend adding some data about the effect of Terminalia bentzoë and etoposide on G0/G1 phase related markers.
- The authors should mention on the physiological relevance of the experimental concentrations of polyphenols used in the study and whether such concentrations could become clinically achievable.
Author Response

(The authors gave the same response as above.)

Reviewer 3 Report
I recommend accepting the manuscript for publication as it is written.
The manuscript is well written. Results are logically sorted and clearly presented. The amount of data is more than satisfactory. From seven plants tested for antioxidant properties and antiproliferative effects, the one most promising, Terminalia bentzoë, was further analysed for genotoxic effects, apoptosis and cell cycle progression. To overcome the problem of complex interplay of numerous secondary metabolites in the crude extract and to characterise bioactive components of Terminalia bentzoë leaf extract more in detail, purification and analyse by extensive spectroscopic methods. Ten phenolic compounds were identified as and tested for cytotoxic activity against HepG2 and for antioxidant activity.
Author Response
Thank you for your recommendation.
Reviewer 4 Report
The manuscript “Focus on Mascarenes endemic plants with specific phytochemical composition, potent antioxidant and antiproliferative properties” by Rummun et al. investigated the potential of plant extracts for therapeutic uses as antioxidants and antiproliferative agents.
The authors performed a complete work assessing the in vitro antioxidants capacity of several plants to continue evaluating the antiproliferative properties of Terminalia bentzoë in different e cell lines, which was in deep evaluated in HepG2 cells (hepatocarcinoma cells).
The authors followed a well-designed protocol to finally identify the compounds exerting the cytotoxic effects on HepG2 using several analytical technics.
In general, the authors performed a good work, but some changes should be amended.
Hence, this manuscript is not ready to be published in Pharmaceuticals by MDPI.
The reviewer comments are listed below:
- Could the title be more specific? As it is, it seems more a review paper than an article.
- The authors should include several sub-sections to explain the antioxidant methods (2.3. In vitro antioxidant capacities of extracts)
- Replace in-vitro for in vitro. The hyphen is not needed.
- Be sure of using mL instead of ml.
- Could the authors quantify the percentage of healthy, apoptotic, and necrotic cells?
- The authors should include an explanation of why they used etoposide as a control.
- In general, figure quality is low. The authors could present more visual figures and follow a continuous style.
- Once the authors separate the fractions, why they did not evaluate the same array of spectrophotometric methods as they did first to select the best plant?
- Thy authors should also be uniform when writing about the significant p-values. p < 0.05, p < 0.01, p < 0.001 etc. is the preferred style. Avoid capital letters and italicize the “p”.
Round 2
Reviewer 1 Report
The authors answered nicely on the comments, but mainly by literature search and addition of literature references to the MS. However, this MS is not a literature review, and as a research paper I still think that the novelty of the MS is rather low, as the identity of the active compounds and their mode of action are not brought here. As said before, this MS is premature. I certainly agree with the authors saying: "this study provided the baseline data for future in-depth investigations of the chemotherapeutic potential of T. bentzoë plant extracts. "
Reasons for publication as brought by the authors "given that these plants are highly threatened these data have their inherent validity for reference purposes" are important, but are more relevant, I believe, to an ecological or medicinal plants-type of journal.
Author Response
Thank you for your comments. This manuscript is submitted for a special issue titled "Anticancer Compounds in Medicinal Plants" in Pharmaceuticals, which aligns with your recommendation.
Reviewer 2 Report
The authors have satisfactorily responded to the comments that the subject matter of this work is acceptable for publication.
Author Response
Thank you for recommendation of acceptance.
Reviewer 4 Report
The authors have appropriately answered the reviewers' comments.
Despite the limitations of the manuscript, the manuscript provides new evidence on the therapeutic use of Terminalia bentzoë (among the other screened plants). The in vitro results could be the first step for further research.
I would have like to see improved quality figures (bar graphs). Excel-based graphs seem to be quite obsolete. The visuals of a paper can favor the public to read it and, therefore, cite it. Hence, as a minor revision, I ask the authors to change their figures, trying to polish them as much as possible.
Take these papers as an example:
Figures made with Excel:
https://www.mdpi.com/1424-8247/13/10/283
Figures made with other software:
https://www.mdpi.com/1424-8247/13/10/271
https://www.mdpi.com/1424-8247/13/10/270
https://www.mdpi.com/1424-8247/13/10/266
https://www.mdpi.com/1424-8247/13/10/266
Author Response
All the Figures previously made by Excel have been remade using GraphPad software with much better quality indeed.
Many thanks!
This manuscript is a resubmission of an earlier submission. The following is a list of the peer review reports and author responses from that submission.